# Sartans and ACE Inhibitors: Mortality in Patients Hospitalized with COVID-19. Retrospective Study in Patients on Long-Term Treatment Who Died in the Italian Hospitals of Area Vasta n.5—Marche Region

**DOI:** 10.3390/jcm11092580

**Published:** 2022-05-05

**Authors:** Tony Mazzoni, Zaira Maraia, Benedetta Ruggeri, Carlo Polidori, Maria Vittoria Micioni Di Bonaventura, Laura Armillei, Irene Pomilio, Isidoro Mazzoni

**Affiliations:** 1Pharmacology Unit, School of Pharmacy, University of Camerino, 62032 Camerino, Italy; tony.mazzoni@studenti.unicam.it (T.M.); carlo.polidori@unicam.it (C.P.); mariavittoria.micioni@unicam.it (M.V.M.D.B.); 2ASUR Marche AV5, 63100 Ascoli Piceno, Italy; zaira.maraia@sanita.marche.it (Z.M.); benedetta.ruggeri@sanita.marche.it (B.R.); laura.armillei@sanita.marche.it (L.A.); irene.pomilio@sanita.marche.it (I.P.)

**Keywords:** ACE-Inhibitors, Sartans, SARS-CoV-2, COVID-19, mortality

## Abstract

Introduction: During the 2019 Coronavirus pandemic (COVID-19), a concern emerged regarding a possible correlation between the severe form of SARS-CoV-2 infection and administration of ACE-Inhibitors (ACE-I) and Sartans (ARB), since long-term use of these drugs may potentially result in an adaptive response with up-regulation of the ACE 2 receptor. Given the crucial role of ACE2, being the main target for virus entry into the cell, the potential consequences of ACE2 up-regulation have been a source of debate. The aim of this retrospective cohort study on COVID-19-positive patients who died is to investigate whether previous long-term exposure to ACE-I and/or ARB was associated with higher mortality due to COVID-19 infection, compared to all other types of drug treatment. Methods: We analysed the clinical and demographic data of 615 patients hospitalized for COVID-19 at the two hospitals of the Vasta Area n.5, between March 2020 and April 2021. Among them, 86 patients, treated with ACE-Is and/0 ARBs for about 12 months, died during hospitalization following a diagnosis of acute respiratory failure. Several quantitative and qualitative variables were recorded for all patients by reading their medical records. Results: The logistic model showed that the variables that increase mortality are age and comorbid diseases. There were no demonstrable mortality effects with ACE-I and ARB intake. Conclusions: The apparent increase in morbidity in patients with COVID-19 who received long-term treatment with ACE-I or ARB is not due to the drugs themselves, but to the conditions associated with their use.

## 1. Introduction

The Coronavirus Disease 19 (COVID-19) infection is a global pandemic caused by the Severe Acute Respiratory Syndrome Coronavirus-2 (SARS-CoV-2). In the Marche Region from 2019 to 2021, COVID-19 infected 112,470 people, causing the death of 3061 people. SARS-CoV-2 is genetically very similar to SARS-CoV-1, the virus that caused the epidemic in 2003, and both microorganisms use angiotensin-converting enzyme-2 (ACE-2) receptors for entry into the body’s cells [1]. The viral Spike protein, also called the S protein (target of antibodies produced by vaccines), located on the outer surface of the virus binds to the ACE-2 receptor. Binding appears to take place between residues 272 and 537 of the viral S protein [2]. The entry of the virus into the cell via the ACE-2 receptor would be much slower and more difficult if it were not ‘helped’ by certain proteases. Several studies suggest that certain proteases located on the cell surface, in very close proximity to the ACE-2 receptor, facilitate virus entry into the cell. In particular, the serine protease TMPRSS2 (transmembrane protease serine 2, a member of the Hep-sin-TMPRSS subfamily) is a transmembrane proteolytic enzyme, which structurally and functionally is part of the ACE-2 receptor (although it is stoichiometrically separated from the enzymatic site of the receptor itself). It is TMPRSS2 that ‘attacks’ the S1 unit of the viral S protein and, through its enzymatic activity, detaches it from the S2 unit. After detachment, the viral S2 unit fuses with the cell and the transfer of the viral content into the cell takes place via this unit [2,3,4]. This enzymatic activity (detachment of the S1 unit) increases virus entry into the cell via the ACE-2 receptor by a factor of almost 100. Based on this, it can be assumed that increased expression of surface proteases within the ACE-2 receptor in the pulmonary alveolar area could help explain the virus’ greater tendency to cause severe bronchioloalveolar infections compared to similar virus strains. 

### 1.1. Up-Regulation of the ACE-2 Receptor in Patients Treated with ACE-I and ARBs

The use of ACE-I and angiotensin II receptor blockers (ARBs) may increase the expression of ACE-2 receptors, thereby increasing vulnerability to SARS-CoV infection [5]. Other authors, however, suggest that these drugs may confer a protective effect, citing evidence that ACE-2 would reduce acute lung injury (ALI) [6]. ACE-2 is expressed in many systems, such as endothelial cells of the lung and the vascular system. [7]. ACE-I and ARB drugs are essential for the treatment of hypertension, chronic kidney disease, heart failure and myocardial infarction. In this clinical setting, strong concern has been raised about a potential detrimental effect of ACE-I and ARB, but similarly abrupt discontinuation of these drugs has been linked to worsening heart failure, dilated cardiomyopathy, destabilization of blood pressure control and increased mortality rates. During the first wave of the pandemic in Italy in March 2020, in the compilation of its weekly reports on the characteristics of patients who died, the Istituto Superiore di Sanità (ISS) drew attention to the use of ACE-I and ARB drugs, highlighting the problem of treatment interruption. The conclusion still indicated by the ISS is the following: “the current evidence does not support the interruption of drug treatment with ACE-I and ARB or the switch to other antihypertensives; therefore, treatment should be continued, as stressed by the various scientific societies and regulatory agencies”. The same problem, whether or not to suspend treatment, has been at the centre of numerous requests from hospital and territorial doctors in the Area Vasta n.5 to the local Pharmacovigilance Centre, since they represent the most widely used drugs in the treatment of hypertension and heart failure. The hypothesis that ACE-I and ARBs may increase patients’ vulnerability to SARS-CoV-2 infection is based on experimental models. Exposure to both ACE-I and ARBs increased ACE-2 expression in the kidney, myocardium and vessels of the rat [8,9,10]. In humans, it was observed that ACE-2 mRNA expression was higher in the myocardium and intestinal tissues of ACE-I users, but not in patients taking ARBs [11,12,13]. In contrast, urinary ACE-2 levels were elevated in ARB users, but not in patients taking ACE-I [14].

### 1.2. Down-Regulation and Role of ACE-2 in Viral Infections

Studies with SARS-CoV and MERS-CoV have shown that, as a result of virus binding, the ACE-2 receptor is down-regulated [15]. This is a biological phenomenon common to various situations of receptor stimulation (e.g., stimulation of beta-adrenergic receptors), which induces down-regulation of the same receptors. In other words, the ACE-2 receptor ‘attacked’ by the virus internalizes (we might say ‘locks itself in’ inside the cell) and thus almost stops ‘working’ [16]. The implications of ACE-2 down-regulation are potentially important. A non-functioning ACE-2 receptor, or a nearly non-functioning ACE-2 receptor, implies a reduced transformation of angiotensin II into angiotensin1–7. As is well known, the ACE-2 enzyme is a dipeptidyl-carboxypeptidase that breaks the bond between proline and the carboxy-terminal phenylalanine’s residue of angiotensin II, consequently transforming angiotensin II (8 amino acids) into angiotensin1–7 (7 amino acids). On the other hand, ACE breaks the bond between phenylalanine and histidine of angiotensin I, therefore transforming angiotensin I (10 amino acids) into angiotensin II (8 amino acids). Angiotensin1–7 has completely “opposite” effects (vasodilation, antiproliferative and anti-fibrotic effects) to those of angiotensin II at angiotensin II type 1 receptors (AT1 receptors). It is clear that a down-regulation of ACE-2 receptors would imply a lower synthesis of angiotensin1–7 and, in parallel, a greater availability of angiotensin II for binding to AT1 receptors. In order to understand the importance of the degradation of angiotensin II by ACE2 receptors, it is essential to review the biological effects of angiotensin II. Angiotensin II is not only a potent vasoconstrictor and stimulant of aldosterone release but has also been shown to be capable of causing adverse reactions such as endothelial dysfunction, myocardial hypertrophy, oxidative stress and increased coagulation. In the lungs, the downregulation of ACE2 receptors would facilitate the progression of the inflammatory and hypercoagulation processes exerted by angiotensin II, which is insufficiently counteracted by angiotensin1–7 [17]. Indeed, a study carried out in mice with acute respiratory distress syndrome induced by sepsis or acid aspiration clearly demonstrated that ACE-2 receptors, as well as angiotensin II type 2 receptors (AT2 receptors), protect these animals from lung injury (pulmonary oedema and reduced function) [18]. This study also showed that ARBs, by blocking AT1 receptors, reduce lung injury induced by sepsis or acid aspiration [18]. Furthermore, the administration of recombinant human ACE-2 attenuated lung injury [18]. Another important study in laboratory animals showed that SARS-CoV, via the S protein, clearly reduces the expression of the ACE-2 enzyme, which increases angiotensin II and worsens lung lesions [15]. These were significantly attenuated by the administration of ARB [15]. An Italian group has shown that angiotensin1–7, whose synthesis is blocked by down-regulation of ACE-2 receptors, significantly reduces inflammatory lung lesions and subsequent fibrosis in experimental models of lung damage. The same group also demonstrated in an animal model that the administration of angiotensin1–7 reduces the diaphragmatic damage (contractile dysfunction and atrophy) commonly observed during mechanical ventilation [19]. Overall, these data suggest that the downregulation of ACE-2 receptors due to virus contact induces a reduction in angiotensin1–7 and an increase in angiotensin II, with consequent deleterious pro-inflammatory effects, particularly in the lungs. We now know that age and the presence of comorbidities are important risk factors for the progression of COVID-19 disease into severe forms. Some conditions, such as heart failure and/or hypertension, are associated with dysregulation of the renin-angiotensin-aldosterone system (RAAS) and ACE2 deficiency. Downregulation of the ACE2 receptor is associated with alveolar wall thickening, edema and recruitment of inflammatory cells. These clinical events underlie the pathophysiological mechanism of SARS-CoV-2 [20]. 

The aim of this retrospective cohort study on COVID-positive patients that died in the hospitals of Area Vasta n.5 of the Marche Region is to investigate whether previous long-term exposure to ACE-I and/or ARB was associated with an increased mortality due to COVID-19 infection, compared to other drug treatments.

## 2. Materials and Methods

We analyzed the clinical, demographic and pharmacological data of 615 hospitalized patients, COVID-19 positive, detected by the SARS-CoV-2 Real-time polymerase chain reaction (RT-PCR) laboratory test. All the patients studied were resident in the province of Ascoli Piceno, corresponding to the territory of Area Vasta n.5, Marche Region. The study population was characterized by 164 patients with SARS-CoV-2 who died during hospitalization from March 2020 to April 2021 with the diagnosis of death: “acute respiratory failure in SARS-CoV-2 positive patient”. COVID-19 positive patients who died of other causes (e.g., stroke, heart attack) were excluded. Among the deceased, 86 patients had been treated with ACE-Is and/or ARBs for 12 months. The clinical data were extracted from the “Primary Care” database available at the hospitals and territorial services of the Ascoli Piceno and San Benedetto del Tronto Vasta Area n.5. In addition to the medical records, “Primary Care” also indicated that the patient was positive for SARS-CoV-2 by RT-PCR. The pharmaceutical data relating to the use of drugs were extracted from the Apoteke Gold management system, which contains the records of all prescriptions over the last 10 years. Apoteke Gold also allows the data of conventionalized, distributed on account or direct pharmaceutical prescriptions to be processed, divided or aggregated, allowing routine epidemiological drug analysis, using the classification system Anatomical Therapeutic Chemical (ATC). Using the patients’ tax code, we extracted the pharmacological treatments from 1 January 2019 until the date of death. Additionally, through the pathology exemption code, we identified previous diseases. The data can be interlinked individually by identifying them with their tax code. The data were analysed by means of two types of statistical analysis: uni- and multivariate analyses. Firstly, descriptive analysis was carried out in order to give a concise representation of the results of the observations. For quantitative variables, counts and percentages were reported. To compare the distribution of the variables between the groups, the chi-square test was used. Finally, multivariate logistic regression was used to estimate the probability of death as a function of the other observed variables (age, the presence of one, two or three concomitant diseases, sex, use of ACE-I and ARB). Only coefficients with a *p* value below the 5% significance threshold were considered statistically significant. Statistical analyses were carried out using STATA software (Stata Corp. College Station, TX, USA).

## 3. Results

During the pandemic, 615 patients were admitted to the hospitals of Vasta Area n.5 with a diagnosis of COVID-19 pneumonia (principal diagnosis) or COVID-19 respiratory distress syndrome (secondary diagnosis). The demographic and clinical characteristics of the study sample are shown in Table 1. We stratified the severity of the disease by subdividing the patients according to the department of hospital admission.

As far as comorbidities are concerned, the sample is very heterogeneous and therefore includes a wide range of diseases. In Table 2, it is possible to observe that in the majority of cases at least one pathology was present in comorbidity. It is also interesting to note that those affecting the cardiovascular system were the most prevalent in the population under analysis (Table 2).

By means of a multivariate logistic regression model, it was possible to identify which variables were correlated with an increased probability of death following SARS-CoV-2 infection. Using the linear logistic model, considering death as dependent factor and as explanatory variable age, the presence of one, two or three concomitant pathologies, gender and therapies, results are strongly dependent on presence of concomitant diseases and age. Therapies and sex do not explain the dependent response. In Table 3, it can be seen that those patients with two concomitant diseases are 2.5 times more likely to die (*p* = 0.006) and those with three concomitant diseases are 6.6 times more likely to die (*p* = 0.000). The model also shows that as age increases, the probability of dying increases by 1.08 (*p* = 0.000). Treatment, sex and a comorbid condition are irrelevant because the *p* value is not statistically significant.

## 4. Discussion

At the beginning of the pandemic, there was concern that the use of ACE-I and ARBs could be harmful in patients with COVID-19, because these long-term drugs have the potential to increase ACE-2 receptor expression and thus lead to a greater likelihood of progression to severe COVID-19 and death [21]. Experimental evidence that ACE-Is and ARBs increase the expression of ACE-2 comes from preclinical studies. Ferrario et al. observed an up-regulation of ACE2 in the cardiac tissue of Lewis rats, while Soler et al. found an increased expression of ACE-2 in renal arterioles [22,23]. In addition to the up-regulation of ACE2, another concern for the use of these drugs in patients with COVID-19 relates to the potential increase in levels of bradykinin, which is knows to be metabolized by ACE. Bradykinin could contribute to the exacerbation of the inflammatory phase by increasing vascular permeability [21]. Data on the outcome of patients who died in the hospitals of Vasta Area n.5 showed a higher mortality rate among users of RAAS inhibitors than among all decedents. However, when other variables were taken into account, it became apparent that the main determinants of mortality might be the older age (*p* = 0.000) and the presence of two or three comorbid conditions (*p* = 0.006; *p* = 0.000), rather than RAAS inhibition itself. In agreement with recent positions taken by many scientific societies, including the European Society of Cardiology, the American Heart Association, the Italian Society of Cardiology, the Italian Society of Hypertension and the Italian Society of Pharmacology, it is possible to speculate that the discontinuation, even temporarily, of ACE-I or ARB in all patients, who are taking them to prevent possible future deaths from SARS-CoV-2, is not supported by convincing scientific evidence. The presence of previous chronic diseases influences the prognosis in people with COVID-19. However, it is not only chronic respiratory disease that determines progression to worse outcomes, but also comorbid diseases of other organs. COVID-19 is a condition that also affects the endothelium of pulmonary vessels, triggering thromboembolic events. Pulmonary inflammation can become systemic and involve other organs (heart, brain, kidneys) leading to multi-organ dysfunction and death. In this context, patients with cardiovascular diseases, diabetes and obesity are more vulnerable and the probability of death from COVID-19 increases among them.

## 5. Conclusions

The results of our epidemiological analysis confirm that there is no correlation between COVID-19 mortality and treatment with ACE inhibitors or ARBs. The data presented above show that the higher mortality rate in patients treated with RAAS inhibitors is due to the presence of diseases for which these drugs are indicated. Our data support the recommendations of several scientific societies to continue these treatments in all patients and confirm the indications of the Pharmacovigilance Office of Area Vasta 5 not to suspend treatment in patients with COVID-19 infection. One of the main limitations of our analysis is the scale of the study; this study was conducted as a double-center study with a limited sample size, despite the apparently higher mortality rate in treated patients. However, this work can contribute, in agreement with other findings produced globally, to eliminate the empirical concern of clinical practitioners. Nevertheless, the strength of the present analysis includes the presence of patients treated for more than 12 months with these drugs compared to previous studies.

## Figures and Tables

**Table 1 jcm-11-02580-t001:** Characteristics of the patients under study.

	Total Patients (*n* = 615)	Deceased Untreated (*n* = 78)	Deceased Treated (*n* = 86)
**Average age**	70.9	68.2	74.6
**Range age**	31–100	31–100	46–97
**Male** **Average age**	366 (59.5%) 68.9	39 (43.3%) 77.7	51 (56.7%) 80
**Female** **Average age**	249 (40.5%) 73.7	39 (52.7%) 81.4	35 (47.3) 83.8
**Intensive therapy**	96 (15.6%)	20 (25.6%)	26 (30.2%)
**Semi-intensive therapy**	128 (20.8%)	13 (16.7%)	25 (29.1%)
**Other ordinary department**	391 (63.6%)	45 (57.7%)	35 (40.7%)

**Table 2 jcm-11-02580-t002:** Comorbid diseases in study sample.

	Total Patients (*n* = 615)	Deceased Untreated (*n* = 78)	Deceased Treated (*n* = 86)	*p*
**No concomitant disease**	140 (22.8%)	8 (10.3%)	7 (8.1%)	0.076
**One concomitant disease**	239 (38.9%)	32 (41%)	2 (2.3%)	0.263
**Two concomitant diseases**	172 (28%)	24 (31%)	34 (39.5%)	0.227
**Three concomitant diseases**	64 (39.5%)	14 (18%)	23 (26.7%)	0.456
**Diabetes**	107 (17.4%)	10 (13%)	20 (23.3%)	
**Hypertension**	86 (14%)	8 (10.3%)	6 (7%)	
**Heart failure**	59 (9.6%)	7 (9%)	27 (31.4%)	
**Atrial fibrillation**	54 (8.8%)	12 (15.4%)	12 (14%)	
**Obesity**	38 (6.2%)	5 (6.4%)	11 (12.8%)	
**Renal failure**	15 (2.4%)	6 (7.7%)	4 (4.6%)	
**Chronic obstructive pulmonary disease**	25 (4.1%)	1 (1.3%)	0	

**Table 3 jcm-11-02580-t003:** Logistic regression analysis.

Variables	Odd Ratio	Std. Err.	z	*p*	95% Conf.	Interval
**One concomitant disease**	1.71	0.57	1.61	0.108	0.88	3.31
**Two concomitant diseases**	2.54	0.86	2.73	0.006	1.30	4.96
**Three concomitant diseases**	6.61	2.65	4.71	0.000	3.01	14.52
**Age**	1.08	0.01	8.04	0.000	1.06	1.11
**Therapy**	1.07	0.23	0.31	0.755	0.70	1.62
**Sex**	0.95	0.20	−0.22	0.826	0.63	1.44

Std. Err.: Standard Error; Conf.: Confidence.

## Data Availability

Data and material are available from the corresponding author.

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
