# Peer review of "Sartans and ACE Inhibitors: Mortality in Patients Hospitalized with COVID-19. Retrospective Study in Patients on Long-Term Treatment Who Died in the Italian Hospitals of Area Vasta n.5—Marche Region"

_jcm, 2022, doi:10.3390/jcm11092580_

Round 1

Reviewer 1 Report

The manuscript has been improved by Authors. No further comments to authors.

Author Response

Thank you very much for your attention. 

Reviewer 2 Report

In this study Mazzoni et al perform a retrospective study of the effect of Sartans and ACE-inhibitors on mortality in COVID-19. The authors conclude:

“Present findings confirm that there is no association between COVID-19 mortality and treatment with ACE inhibitors or ARBs and show that the apparently higher mortality rate among RAAS inhibitor users is not due to the drugs themselves, but to the conditions associated with their use.”

Although this reviewer appreciates the efforts of the authors this study confers very little new data to the field. As stated in the manuscript many more larger studies have already investigated this topic and come to the same conclusion. In addition, there may be two reasons for not finding an effect of the drugs. The first is that there is no increased mortality due to Sartans and ACE-inhibitors (which is probably correct but cannot be concluded with this study design). The other reason is that there is not enough power in the study to detect such a difference. The reason here is not known, thus the authors cannot draw the conclusion that ACE inhibitors or ARBs do not cause a higher mortality rate in COVID-19.

Author Response

Dear Reviewer, we would like to thank you for your comments.

As mentioned in the conclusion, one of the main limitations is the scale of the study. As a double centre study with a limited sample size, we do not claim to define the outcome of therapy, but we hoped that our data would be in agreement with those produced globally. In addition, patients were treated with these drugs for more than 12 months before hospitalisation. This study considered patients who had died compared to the other studies.

What allowed us to state the non-correlation between COVID-19 mortality and the use of ACE-Is and/or ARBs was the multivariate logistic model (table 3 in the article). This model was calculated by taking into account all patients (n=615) and all variables involved (sex, age, disease and treatment) in order to get a complete overview. Although treated patients had a probability of dying 1.07 higher than untreated patients, the p value was not statistically significant, but was statistically significant in the case of the age variable (p=0.000). In addition, the model showed that those with two concomitant diseases were 2.5 times more likely to die (p=0.006), and those with three concomitant diseases were 6.6 times more likely to die (p=0.000). We have discussed these results in the paper. The logistic model has the power to consider all variables simultaneously by adjusting the estimate between them.

Come suggerito, il disegno dello studio è stato migliorato e l'articolo è stato reso più facile da leggere. L'inglese è stato rivisto da un madrelingua. 

I nostri dati potrebbero potenzialmente supportare e contribuire alla letteratura scientifica con l'ultima speranza di eliminare la preoccupazione empirica che esiste ancora tra i professionisti clinici.

In attesa di una risposta,

Ti ringraziamo,

Distinti saluti.